# Artificial Humic Substances as Biomimetics of Natural Analogues: Production, Characteristics and Preferences Regarding Their Use

**DOI:** 10.3390/biomimetics8080613

**Published:** 2023-12-15

**Authors:** Elena Efremenko, Nikolay Stepanov, Olga Senko, Ilya Lyagin, Olga Maslova, Aysel Aslanli

**Affiliations:** Faculty of Chemistry, Lomonosov Moscow State University, Lenin Hills 1/3, Moscow 119991, Russia; na.stepanov@gmail.com (N.S.); senkoov@gmail.com (O.S.);

**Keywords:** waste treatment, composting, anaerobic digestion, pyrolysis, hydrothermal carbonization, biosorbents

## Abstract

Various processes designed for the humification (HF) of animal husbandry wastes, primarily bird droppings, reduce their volumes, solve environmental problems, and make it possible to obtain products with artificially formed humic substances (HSs) as analogues of natural HSs, usually extracted from fossil sources (coal and peat). This review studies the main characteristics of various biological and physicochemical methods of the HF of animal wastes (composting, anaerobic digestion, pyrolysis, hydrothermal carbonation, acid or alkaline hydrolysis, and subcritical water extraction). A comparative analysis of the HF rates and HS yields in these processes, the characteristics of the resulting artificial HSs (humification index, polymerization index, degree of aromaticity, etc.) was carried out. The main factors (additives, process conditions, waste pretreatment, etc.) that can increase the efficiency of HF and affect the properties of HSs are highlighted. Based on the results of chemical composition analysis, the main trends and preferences with regard to the use of HF products as complex biomimetics are discussed.

## 1. Introduction

Humification (HF) is the natural process of converting bioorganic matter into humic substances (humus, humate, humic acid, fulvic acid, and humin) via geo-microbiological mechanisms under aerobic and/or anaerobic conditions [1,2,3,4]. Humic substances (HSs) and their composition and concentrations mostly determine the basic properties of soils and play an important role in regulating the growth of plants and soil microorganisms [5,6] and the accumulation and migration of metal ions, radionuclides, and ecotoxicants in soils [7]. It is possible to regulate these processes by changing primarily the concentration of HSs in soils due to their introduction [6]. Actively used and damaged soils necessitate the constant introduction of HS in significant quantities in order to restore them [8], while the presence of main the components of HSs (humic acids (HAs) and fulvic acids (FAs)) and their quantitative ratios predetermine the functions of HSs as soil structurators involved in the regulation of soil humidity and air and water permeability. Currently, the raw materials for the commercial production and widespread use of HSs are mainly peat and coal, from which HSs are extracted [9,10]. In nature, the formation of HSs occurs as a result of the HF of bioorganic (mainly plant) residues. However, various methods similar to natural ones are being developed for the “artificial” production of HSs from organic wastes accumulated mainly in agriculture (plant waste and animal excrement). These wastes are generated annually in significant quantities on livestock and poultry farms in various countries around the world (Table 1) [11,12,13,14,15,16,17,18,19,20,21,22,23]. Their conversion into HS resources containing simultaneous sources of carbon (C), nitrogen (N), phosphorus (P), and microelements allows us to simultaneously solve the problem of obtaining HSs similar to natural ones and significantly reducing the volumes of waste. In addition, a number of methods for the HF of these wastes allow, together with the production of HSs, for the destruction of various micro-pollutants contained in them (pesticides, mycotoxins, microplastics, pharmaceutical pollutants, etc.) [24,25,26,27]. However, the initial composition of wastes and the applied HF methods not only lead to the acquisition of various products (Figure 1) but also HSs with different compositions and properties, widening the range of fields of their potential use instead of natural HSs.

The purpose of this review is to analyze current information on various HF methods for animal wastes (AWs) resulting in HS accumulation to compare the content of HSs depending on the sources and HF process conditions to discuss the prospects of the use of HS resources from AW instead of traditional natural HSs from peat and coal.

## 2. Different Methods Used for Artificial HF of Animal Wastes (AWs)

Various biological (anaerobic digestion (AD) and composting) and physical–chemical (hydrothermal carbonization (HTC), pyrolysis, acid or alkaline hydrolysis, and subcritical water extraction) methods for the HF of AW are currently being investigated (Figure 1, Table 2 [28,29,30,31,32,33,34,35,36,37,38,39,40,41,42,43,44,45,46,47,48,49,50,51,52,53,54,55,56,57,58,59,60,61,62,63,64,65,66,67,68,69,70,71,72,73,74]). To compare the processes and HS resources obtained using them, the following indicators were taken into account in this review: HF ratio (the ratio of all HSs (HA plus FA) to the total carbon content (TOC) in the resulting product); the HF index (the ratio of certain acids (HA or FA) to TOC); the percentage of HA or FA in the product (the ratio of the HA or FA concentration to the total concentration of HSs in the product); HS yield (the weight ratio of the obtained HSs to the processed substrate, which can be represented as a percentage); and the polymerization index (the ratio of HA/FA) [75].

### 2.1. HF via Composting of AW

Composting with the formation of compost (Figure 1) containing different concentrations of HSs (26–260 g/kg) as the main product is widely used among the biological methods for the HF of AW (Table 2). Composting is an aerobic process during which the microbiological degradation of bioorganic substances occurs along with the formation of HS precursors (amino acids, reducing sugars, peptides, etc.) with their subsequent stabilization in self-forming supramolecular ensembles [76]. Composting consists of several stages: heating, cooling, and maturation. The temperature in the composted mass begins to rise a few hours after the beginning of the process as a result of biochemical degradation reactions with an increase in the concentrations of reducing sugars, nucleic and amino acids, phenol residues (in the case of the conversion of lignin-containing compounds present in animal excrement), etc. The subsequent decrease in the concentrations of these compounds occurs due to the formation of structurally complex HSs at the compost maturation stage [36].

The formation of HSs during composting is influenced by many factors (temperature, pH, C/N ratio, humidity, oxygen concentration, etc.). The maturity of compost is determined precisely by the change in the HS concentrations in its content [28,31,33,35,45]. An increase in the composting temperature ameliorates the destruction depth of organic compounds and promotes their interaction with the formation of HSs. The dynamics of increasing HS concentrations in compost also depend on temperature. The concentration of HA remains relatively constant during the first 20 days of the process and then gradually increases under mesophilic composting conditions [28]. With an increase in the composting temperature (up to 90 °C), the HS concentrations decrease by 25% in the first 30 days and then rise sharply as a result of the formation of HSs concurrently with the cooling and maturation of the compost [38,40,43]. There are general trends in the simultaneous increase in HA content and decrease in FA concentrations in the HS composition during mature compost formation [35,43].

FAs can be used as substrates by microorganisms and converted into HA via condensation and polymerization reactions, which lead to an increase in the aromaticity of HSs and an increase in HF. The HF index is considered to rise as the ratio of HA/FA increases [30]. The HA/FA ratio can reach 5.4–7.6, as a rule, after 1.5 months of composting in the final compost of AW [43]. The average HS concentration can reach 90–100 g/kg in mature compost with a predominance of high-molecular HAs in the HS composition, although the overall HS concentrations can vary in the compost due to the different initial characteristics of the treated AW (Table 2).

The sequence of transformations of various functional groups in HS formation in a compost was analyzed using two-dimensional correlation spectroscopy (2D-COS) [34,35]. First, the C–O stretching of aromatic acid and aliphatic acid esters was observed. Then, the C–H deformation, vibration, and C–O stretching of polysaccharides or polysaccharide-like substances was observed. Further, the C=O stretching of carboxylate, quinone, ketone, or amide was witnessed. Finally, the C=C stretching of aromatic rings was confirmed [34]. The elemental composition of HSs during the composting of AW under hyper thermophilic conditions showed an increase in N-content, which is associated with enhanced polymerization and condensation of polysaccharide-like substances with N-containing compounds (proteins, nucleic acids, and amino acids) [35].

Composting is a long process: it can span from 1 to 2 months (Table 2) to 6 months or longer (with vermicomposting) [46]. Composting is characterized by low process speeds (Figure 2) in spite of high yields of conversion of organic substances into HSs. This is the reason why alternative methods for the HF of AW are being developed, leading to the production of HSs and the search for factors capable of HF acceleration.

### 2.2. HF via Anaerobic Digestion of AW

Another biological method of converting AW into a product containing HSs is AD (Table 2, Figure 1). This process allows one to simultaneously obtain biogas—consisting mainly of CH_4_ and CO_2_, used as an alternative energy source—and digestate (Figure 1), containing a consortium of methanogenic cells, products of their metabolism, and HS-containing products of the anaerobic degradation of AW. Another composition of microorganisms involved in AD [77], in comparison with composting, determines other rates of HS accumulation in the resulting digestate (Table 2, Figure 2). Due to the action of the microbial consortia, some of the substances initially present in AW do not undergo destruction and conversion or undergo these processes extremely slowly under AD conditions. This applies to the lignocellulose components of processed excrements [78] and affects the characteristics of AD and the resulting product with HSs. The presence of organic N-containing compounds in the treated masses is another problem in the HF of AW via AD as compared to composting.

To reduce the effect of N-containing compounds (in particular, urea) on AD, their membrane separation from the reaction medium is considered during the process [21]. For the successful conversion of AW into digestate containing HSs, the control of a larger number of factors in comparison with composting is required. The initial content of AW loaded in the AD reactor, the current pH value of the fermentation medium, temperature, the concentrations of the products formed, pressure created by accumulated gases in the working AD reactor, etc., should be controlled [25].

As in composting, with an increasing duration of the process, the concentration of humic-like compounds (HLC) gradually increases, while, on the contrary, the concentration of fulvic-like compounds (FLC) decreases [47]. These are common trends in changes in the composition of HSs in products obtained in two different biological processes for the HF of AW (composting and AD). The use of 2D-COS [48], as in the case of composting, made it possible to study the structural changes in the active functional groups of HSs obtained as a result of the AD of chicken manure. It was found that the active functional groups of HSs changed in the following sequence: aliphatic-like substances (C–H), amides (H–N) or carbohydrates (O–H), carboxylic acids (C=O), polysaccharides (C=O), aromatic compounds, and ketones (C=C). If we compare these results with those mentioned earlier for the HF of AW via composting (Section 2.1), then we can note a clear general similarity in the identified sequences of changes in organic matter after its HF under aerobic and anaerobic conditions.

During the HF of lignocellulose raw materials, many more HLC are formed in AD than in the similar AD of AW. Lignocellulose provides an increased number of precursors for the formation of HSs, although the HF of plant raw materials under AD conditions is much slower [47].

It has been established that HSs formed with a more complex and stable structure [49,50] from lignocellulose are characterized by a high degree of aromaticity and have a notable inhibitory effect on AD. For the subsequent biodegradation of such HSs, their oxidation is necessary, and under AD conditions, this type of conversion is less possible than in composting. In this regard, AWs are more attractive substrates for the formation of HSs via AD in comparison with lignocellulose raw materials. The accumulation of HSs during AD reduces the velocity and efficiency of the process due to the inhibition of the hydrolytic activity of cells participating in the functioning of methanogenic consortia [77,79]. A decrease in the HS concentrations in the digestate can also be observed with an increase in the temperature of the process due to the predominant formation of gas products (CH_4_ and CO_2_) and not HSs. In this regard, the HF of AW via AD proceeds at lower speeds than that via composting (Figure 2), but since AD is accompanied by the production of biogas, which can be used as biofuel, this process of HS acquisition retains interest.

### 2.3. Hydrothermal Carbonation and Wet Torrefaction of AW

Hydrothermal carbonation (HTC), carried out in an aqueous environment at 170–280 °C and high pressure for several minutes or hours, can be used for the HF of AW at higher speeds than those in composting and AD (Table 2). Initially, a mass of AW is prepared, which is dried, crushed, and then mixed with water in different proportions [52,55,56,57,59]. HTC begins with the hydrolysis of high-molecular-weight compounds to monomers, and then the dehydration of monomers occurs, and the final product (hydrochar) containing HSs (Figure 1) is formed due to polymerization and aromatization processes [59]. Liquid (organic (acetic, propionic, and butanoic) acids [55], ketones, aromatic compounds, aldehydes, and alcohols) and gaseous (mainly CO_2_, CO, and CH_4_) products can accumulate in HTC. Depending on the conditions of the process and the AW used, it is possible to obtain final products with different yields, chemical compositions, and characteristics of the product with HSs [53]. Interestingly, the yield of hydrochar in the HTC of chicken manure (44.6%) was almost close to the result obtained with swine manure (43.4%) [56], while it exceeded the same characteristic of the HTC of lignocellulose raw materials by 4–5%. Among the HSs detected in the composition of hydrochar, 20% were compounds corresponding to HSs isolated from the soil, 30–40% were FLCs with a large molecular size, 10–25% of the compounds were characterized as “reduced quinones” with high aromaticity, and 12–23% were protein-like substances containing structures similar to aromatic amino acids (tyrosine and tryptophan) [58].

Wet torrefaction is a process similar to HTC [63,64] and can be conducted in aqueous or steam-water media at temperatures slightly lower than those for HTC (150–260 °C), with a processing time of up to 40 min (Table 2). The torrefaction temperature has a significant effect on the residual humidity, ash content, and yield of the resulting biochar produced from AW [63]. The yield of hydrochar with HSs decreases with an increasing process temperature [64]. The characteristics of the hydrochar should be carefully controlled. Despite incurring significant energy costs, as well as necessitating the use of special equipment that ensures the maintenance of the necessary temperature conditions of the process, HTC is currently attracting a lot of attention due to the high speeds of HF. The HS yields during the HF of various AWs via HTC (Table 2, Figure 2) are similar to those known for the natural HF of organic matter.

### 2.4. HF of AW by Pyrolysis

Among the physical–chemical methods used for the HF of AW, pyrolysis is one of the most actively studied and consists of the heat treatment of dry raw materials in the absence of oxygen. Pyrolysis results in the formation of biochar with HSs; bio-oil with benzenes, alcohols, alkanes, alkenes, ketones, phenols, and poly aromatic hydrocarbons; and non-condensable gases (CO_2_, CO, CH_4_, H_2_, NH_3_, H_2_S, and hydrocarbons). The yield of biochar with HSs is determined by the conditions of the process and the composition of the AW used (Table 2).

In the pyrolysis of AW, with an increase in temperature (from 400 to 700 °C) and the duration of the process (from 20 to 40 min), an increase in the proportion of the gas fraction among the products obtained (up to 40–60 wt.%) will manifest. The yield of biochar decreases with an increasing temperature (from 38 to 28 wt.%), and the degree of aromaticity of the HSs present in it increases [60]. Usually, up to 50% of the initial content of C, N, and S in the AW is lost in the form of volatile compounds, whereas the relative content of ash and metals increases by 2–2.4 times in comparison with AW [61,66]. The composition of HSs in biochar is similar to the composition of HSs in chernozem and peat, which are characterized by a large number of surface -COOH groups, providing them with a large capacity for cation exchange. The sum of all O-containing functional groups (C–O, C=O and COOH) in the composition of biochar HSs is lower than that of HSs from various types of soils, indicating the higher hydrophobicity of the biochar surface.

In general, the HSs of biochar obtained as a product of the pyrolysis of AW are similar in their production rates and yields to the same parameters known for HTC (Figure 2); however, higher temperatures are required for pyrolysis. In this regard, pyrolysis, as a method for the HF of AW, refers to processes with high energy consumption but involving HSs, which, in their elemental composition and characteristics, are as close as possible to natural analogues coming from coal.

### 2.5. Acid and Alkaline Hydrolysis as a Method for the HF of AW

As alternatives to high-temperature methods for the HF of AW, acid and alkaline hydrolytic processes are being actively investigated (Figure 1, Table 2). The efficiency of the conversion of AW into HSs using hydrolytic treatment depends on the type of processed mass [71]. The use of HCl or H_2_SO_4_ for the acid hydrolysis of AW is very effective for processing mass with a high content of polysaccharides [71]; however, other components of AW remain without effective hydrolytic action. The efficiency of the alkaline hydrolysis of AW depends on the alkaline agent used, for which KOH, NaOH, NH_4_OH, CaO, Ca(OH)_2_, and CaCO_3_ are applied in different concentrations (0.1–2 M) [24,72]. The temperature of the process (24–100 °C) and its duration (1–48 h) affect the extent of the HF of AW. These hydrolytic processes are interesting because they combine hydrolytic reactions, HF, and the accumulation of HSs in the liquid phase, which is usually carried out via the extraction of HSs from natural sources (coal and peat). According to the speed and the applied temperature conditions, hydrolytic processes occupy an intermediate position between biological (composting and AD) and physical–chemical high-temperature methods for the HF of AW (HTC and pyrolysis) (Figure 2). They ensure the immediate production of HSs dissolved in the reaction medium, without incurring significant energy and large time costs in the process. Innovative studies using subcritical water extraction, in which H_2_O and CO_2_ act like organic solvents such as methanol and chloroform, respectively, should be noted in this review [74]. A high yield of HSs from chicken manure (51 g/kg) was achieved during subcritical water extraction at 250 °C and a pressure of 50–60 atm. An increase in the temperature of the process (to 270 °C) led to a decrease in the content of HSs, just like what was noted earlier in other processes of HF. However, so far, such studies are rare, since the process is energy-intensive, requires subcritical extractants, and necessitates the use of expensive equipment that can generate high temperatures and pressure. Thus, alkaline hydrolysis is under active investigation because it enables the acquisition of HSs from AW quite easily, in a relatively short time, and at compromise temperatures. At the same time, the productivity of this process is lower than that of HTC and pyrolysis, and the issue of alkaline solid-phase waste disposal or application after separation of the liquid fraction with HSs remains unresolved.

## 3. Approaches to Intensification of Artificial HF of AW

Using different methods for the artificial HF of AW, especially biological methods, many researchers are searching for approaches to improving the indicators of HF (the speed and depth of the process) (Table 2). The use of various additives for incorporation in the reaction media (lignocellulose containing wastes, mineral particles, biochar, hydrochar, lignite, metal salts and nanoparticles, oxidizing agents, acidic or alkaline agents, conductive and dielectric materials, precursors of HSs, suspended or immobilized microbial cells, artificial consortia, etc.) is mostly studied to improve the characteristics of HF processes. Several procedures for AW pretreatment have also been discussed as efficient solutions for HF improvement. Some of these approaches are common for different methods of HF, but, in some cases, preferences remain.

### 3.1. Composting

Cell immobilization [80], the construction of artificial consortia, and the introduction of precursors of HSs or compounds, capable of accelerating the hydrolysis and oxidation of substrates, into a medium are widely used for the improvement of results yielded by biological methods of HF (Table 2). The addition of cellulose-containing wastes (straw, sawdust, corn stalk and rice hulls, spent mushroom substrate, bagasse and bagasse pith sugarcane leaves, and spent coffee grounds) (Table 2), which sorb substances inhibiting HF via composting (heavy metals, organic acids, etc.) and change the C/N ratio in HSs [81], is effective for introduction into the compostable mass. To improve the HF of various AWs via composting, the application of mineral sorbents (black tourmaline [33], red sludge [41], montmorillonite, and illite [44]) has been reported. The use of various oxidizing agents (H_2_O_2_, MnO_2_, CuCl_2_, Fe_2_O_3_, and Al_2_O_3_) during composting makes it possible to accelerate the decomposition of organic substances, especially lignocellulose in the content of AW, and their conversion into HSs [30,33,82]. A 20% increase in HSs in compost can be achieved in this case. An enzyme such as laccase [32,82], giving rise to H_2_O_2_ in the composted medium, can be successively applied as an oxidizing catalytic bio-additive instead of the direct loading of a chemical reagent (H_2_O_2_). Interesting, enzymes with relatively non-stable biotic catalysts show higher oxidative efficiency than metal oxides in composting, whereas abiotic catalysts can be produced at a lower cost. It can be expected that future developments in this field may be based on using combinations of enzymes with metal-containing catalysts for HF. This approach can simultaneously lead to improvements in the decomposition of AW and the detoxification of treated media contaminated with mycotoxins [83].

The introduction of exogenous precursors of HSs in the form of amino acids [84] and benzoic acid [30] or inhibitors of the tricarboxylic acid cycle (adenosine triphosphate and molonic acid), reducing the formation of CO_2_ [37,38] during AW composting, leads to an increase in the concentration of HSs, especially HA (by 66.9%), in compost. To accelerate composting, AWs are heated to a temperature above 80 °C for several hours [28,35,36]. To reduce carbon loss (up to 77% due to CO_2_ emissions), biochar is introduced into the composted AW [42]. Biochar retains water, maintains desired pH, and can be used by microorganisms as a carrier for immobilization, improving the stability of cell metabolism and the conversion of organic substances into HSs [42]. 

The porous structures of various minerals and their large surface area and ion exchange and adsorption–desorption capacity promote the use of such additives in the composting of AW, improving the metabolic activity of microorganisms, reducing NH_3_ and N_2_O emissions, and enhancing the HF of AW [41,44]. The introduction of cells of various microorganisms [31,40] induces the biotransformation of organic matter in compost and the formation of HSs (Table 2). This approach is most often used in practice, and there are many biologics on the market that improve and accelerate compost maturation. However, obtaining compost with a modified microbial composition and its subsequent use requires the monitoring of the compost for toxicity before use.

### 3.2. Anaerobic Digestion

For the HF of AW via AD as well as via composting, the hydrothermal pretreatment of excrements at 70–170 °C has been confirmed to be effective [51]. This process increases the biodegradability of manure components, increasing the level of FLC formation in the digestate. For the HF of lignocellulose as a part of undigested feed residues, the pretreatment of AW is carried out before AD. Physical–chemical and biological methods for the pretreatment of AW, including acid or alkaline hydrolysis, alkaline hydrogen peroxide pretreatment, hydrothermal or enzymatic treatment, and hydrodynamic cavitation [78,85], have been investigated to improve the AD efficiency of AW. The use of urea as an additive for AD has been reported [86]. This method allows for the maintenance of the pH balance of a medium, neutralizing organic acids accumulated in the digestate and providing favorable conditions for HF. The combined addition of urea and KOH to an AD medium increases the biodegradability of AW and promotes HF [87]. Zeolites, FeSO_4_, MgCl_2_, MgSO_4_, or MgCl_2_ are added to media with AW since high concentrations of N-containing compounds in media for AD are not desirable due to their conversion to NH_3_ [88]. This leads to a 20% increase in the concentration of HSs formed.

An increase in the HS concentration in the digestate inhibits the metabolic activity of cells that catalyze AD. The use of immobilized forms of natural and artificial anaerobic methanogenic consortia instead of suspended analogues reduces the inhibition of cells by HSs [89]. The immobilization of anaerobic cells in high concentrations makes it possible to obtain efficiently functioning biocatalysts in a state of quorum sensing [24]. The use of artificially created anaerobic consortia for the HF of AW makes it possible to increase the rate of AD even in the presence of micropollutants [25,90].

The efficiency of AD can be improved by introducing biochar (Table 2) or conductive and dielectric materials (stainless-steel mesh and carbon or polyester felt) to reaction media [91]. Such materials can improve the direct interspecies electron transfer and act as a carrier for the formation of stable microbial biofilms inside an AD reactor. Studies on new composite additives for the HF of AW via AD appear attractive and constitute a novel trend in the development of science. The addition of various salts (CaCl_2_, MgCl_2_, and FeCl_3_) to the AW for HF via pyrolysis increases the aromaticity and stability of HSs in the resulting biochar [92]. The addition of H_3_PO_4_ and MgO during the pyrolysis of AW in a mixture with coffee husks increases the yield of biochar up to 65% [67].

### 3.3. Hydrothermal Carbonation and Pyrolysis

Interestingly, the addition of acids (CH_3_COOH or H_2_SO_4_) to AW before HTC increases the yield of hydrochar and the concentration of HSs in it [54]. However, the presence of alkaline additives (CaO or NaOH) in the pyrolysis of AW more positively affects the characteristics of biochar by increasing its pH, the aromaticity of HSs, electrical conductivity, and ash content (Table 2). The addition of a similar alkaline agent (KOH or CaO) to HTC media similarly increases the yield and porosity of hydrochar from AW and the content of HLC in it [52,58]. In fact, the combination of pyrolysis or HTC with the alkaline hydrolysis of AW yields improved results in terms of HF and the quality of the resulting products.

## 4. Comparison of Characteristics of Natural and Artificial HSs Obtained through Various Processes for the HF of AW

It is extremely interesting to compare the content of the main chemical elements of HS obtained via different methods for the HF of AW and of HSs from different natural sources (Table 3) [35,46,48,52,53,54,55,56,60,61,62,64,67,68,72,73,93,94,95]. The results of such a comparison should help us to assess the possible and most appropriate ways to use artificial HSs.

The H/C ratio is an important indicator determining the aromaticity of HSs. Higher atomic H/C ratios correspond to lower aromaticity [96]. The high aromaticity of some HSs ensures their stability under environmental conditions. The H/C ratio ranges for HSs obtained via the HF of AW (0.05–1.86) and for HSs from traditional sources (0.06–1.23) are comparable (Table 3). For products obtained via the HF of AW, this indicator depends on the source of wastes and the process used. Thus, HF via AD yields more aliphatic HSs than that via composting, making them more easily biodegradable when applied to soils. In this regard, digestates appear to be more attractive for use as agrochemicals and are comparable in this regard with HSs from peat. Most samples of HSs in products obtained in the processes of HTC and pyrolysis are characterized by increased aromaticity and are close in their characteristics to the HSs from natural coal. 

High C/N ratios of HSs correspond to a high condensation degree and HF degree of organic matter. This ratio is the key for ensuring favorable conditions for the functioning of microorganisms in soils when using HSs as agrochemicals. The optimal value of this parameter is 20–40 [97]. For HSs obtained from traditional sources (peat and coal), the C/N ratio is within the optimal range (Table 3). Among all the methods of HF, pyrolysis and the HTC of AW yield the maximum C/N ratios in the HSs of obtained products. For HS samples produced from poultry manure, the C/N ratio is on average lower than those values revealed for HSs from the wastes of other animals, regardless of the method of HF used, since chicken wastes contain a lot of N (Table 3). Biological methods of HF (composting and AD) and alkaline hydrolysis of AW lead to the formation of HSs with high N content and, accordingly, with a low C/N ratio (Table 3). Such HSs are not promising as agrochemicals because balancing of the C/N ratio is required. However, a C/N ratio of less than 10 leads to the activity inhibition of microorganisms, including pathogenic ones [97]. This can be taken into account and used for the inhibition of negative microbial processes by such HSs.

The O/C ratio in HSs reflects the content of oxygen-containing functional (carboxyl, hydroxyl, and carbonyl) groups in combination with aromatic structures and serves as a testament to the ability of HSs to enter into exchange and donor–acceptor interactions, form hydrogen bonds, and actively participate in sorption processes. The O/C ratio (Table 3) for the majority of HSs obtained by various methods from AW is comparable to the same evaluator for HSs extracted from traditional sources (0.18–0.55). However, the pyrolysis and alkaline hydrolysis of air-dried poultry manure produced HSs with maximum O/C ratios (0.99 and 1.07), determining the area of effective use of these products for sorption processes and the remediation of soils.

The ratio of N/S is important for plants, and the optimal range is 5–15 [98]. The sulfur content in artificial HSs is significantly lower in most cases than in traditional samples of HSs (Table 3), and the high level of N in artificial HSs provides the necessary N/S ratio (4.8–11.7). Natural HSs have an N/S ratio lower than optimum (0.3–3.4). This is evident when comparing HSs in natural coals and HSs in chars obtained via physical–chemical methods of artificial HF from AW. So, this information is useful for considering the possible replacement of natural coals with artificial chars (biomimetics) in their applications (Figure 3).

Additionally, it can be noted that it is currently impossible to unambiguously determine which additives are the best for a particular method for the HF of AW, and there are no universal solutions to the problem of improving the quality of the HSs obtained and the efficiency of the HF processes themselves. This is why such studies remain relevant and continue to be conducted. Perhaps the use of combinations of already known additives can facilitate additional improvements to the discussed characteristics of HSs and processes.

## 5. Prospects and Preferences for the Use of HS-Containing Products Obtained via HF of AW

Samples of HSs from AW can be used in various products (compost, bio- and hydro-carbons, anaerobic digestate, hydrolysates of AW, etc.) (Figure 3, Table 2). The palette of applications can be very wide, and this has already been confirmed in current studies (Table 4) [99,100,101,102,103,104,105,106,107,108,109,110,111,112,113,114,115,116,117]. Analysis of the characteristics of artificial HSs from AW (Table 3) showed that it is possible to expect that they can be used in a similar or more effective manner in comparison with HSs from coal and peat.

The use of hydrochar obtained from AW during HTC leads to an increase in the HS content of the soil, which contributes to an increase in rice yield by 36% [113]. Hydrochar made from the digest of manure is effective as a soil improver for lettuce growth [115]. Biochar obtained via pyrolysis, when added to soil with cattle manure, causes changes in the content of microbial communities, increasing the number of fungi–plant symbionts [112]. Sources of nitrogen (N) are often added to natural HSs, or the HSs themselves are specially modified, for the subsequent slow release of nitrogen compounds into the soil for plant growth [118]. However, in the case of HSs obtained from chicken manure, their N content has already been increased (Table 3), allowing for the use of artificial HSs as organic fertilizer without additional modification and N enrichment.

One of the main properties of chars from AW for agricultural use is their electrical conductivity (EC) (the higher the EC, the higher the concentration of soluble salts). It was shown that the EC of biochar HSs decreased with an increase in the pyrolysis temperature of dairy cattle manure. Low EC values of biochar, used as an agrochemical, reduces soil salinity. In addition, the water holding capacity of biochars obtained at pyrolysis temperatures ranging from 300 to 550 °C was 380–485%. Thus, such biochar can improve the water-holding capacity of soils, and by varying the temperature of biochar pyrolysis, this characteristic can be changed and controlled [119].

It should be emphasized that HSs, naturally accumulating in soils mainly as a result of transformations of decaying lignin-containing plant residues, consist of aliphatic and aromatic (condensed and non-condensed) compounds, and the latter are present in significantly lower concentrations than aliphatic structures. The pyrolysis of lignin-containing organic matter results in a yield of mainly condensed compounds in the content of HSs [120]. The characteristics of biochars obtained via the pyrolysis of AW significantly depend on the scheme of the process, and the ratio of hydrophobic to aromatic compounds can be specifically increased via increases in temperature and the prolongation of the process [121]. The final content of HSs of biochar and their characteristics can be varied through the management of process conditions. It was shown that purposely increasing the hydrophobicity of HSs in AW biochar obtained via pyrolysis improves their effectiveness toward phenanthrene sorption in the soil [122].

HSs are used as ameliorants for the bioremediation of soils contaminated with oil [123], heavy metals [114], and salts [124]. Similarly, hydrochar and biochar obtained through the HF of AW are used not only as agrochemicals but also as adsorbents for the removal of various substances polluting soil and water [57,59,68,125,126]. Biochar obtained via the pyrolysis of pig manure was used for the sorption of uranium, and its treatment with NaOH significantly increased sorption capacity [104]. HSs isolated from composted cow manure were compared with HSs obtained from coal (leonardite) with regard to removing metals (Cu, Pb, Zn, Cd, and Cr) from the soil [107]. HSs from compost had a greater complexing ability than HSs from coal due to the high content of carboxyl and phenolic groups. HSs extracted from vermicompost adsorbed up to 99.7% of aflatoxin B1 from broiler feed [111]. Biochar from pyrolyzed chicken manure was used as an adsorbent to remove phenol and 2,4-dinitrophenol from wastewater [99]. The loss of the adsorption capacity of this biochar was less than 20% after five repeated uses. A hydrochar obtained from pig manure via HTC and modified with MnFe_2_O_4_ nanoparticles was repeatedly used to remove chlortetracycline and Cd from water [102].

The addition of HSs from AW in the form of hydro- or biochar to reaction media for the AD [108] or composting of different wastes [109,113,114] leads to an improvement in their HF indicators. It has been established that for artificial HSs not obtained from AW in processes similar to those discussed above (Table 2 and Table 4), the possibility of their use has already been demonstrated in a number of areas where HSs from AW have not yet been studied. For example, it has been shown that HA (10 g/L) from HSs of compost in combination with KH_2_PO_4_ and H_2_O_2_ can remove up to 90% of diesel pollution from soil [110]. Hydrochar from olive mill waste and cellulose can be used for enzyme immobilization [115] and the production of electrode materials [116]. The antioxidant and antimicrobial activity of HSs extracted from compost is interesting with respect to their use in the treatment of contaminated objects and media [117]. Hydrochars can also be used as catalysts in various processes and for energy production [127], as well as for the capture of free radicals, cholesterol, glucose, and viral particles [128]. The use of HSs as feed additives [9], components of building materials (fresh mortars and aggregates in cementitious composites) [129], various composite materials (wood polypropylene composites, plastics, etc.) [130], and textiles [131] with improved functional properties has been demonstrated.

Thus, HSs obtained from various AWs using different methods of HF can be effectively used in a variety of fields along with natural HSs derived from coal and peat. It is obvious that some of those fields of potential application that are already known for natural HSs are still unexplored regarding artificial HSs from AW. However, the areas and volumes of their possible use present serious commercial and environmental potential.

The analysis of the data, shown in Figure 2 and Table 3, revealed that composting is the most commonly used method for obtaining organic matter, but it is not the best method because it is very long and relatively nonproductive. It seems simple and can be used for the treatment of relatively low amounts of animal waste with good outside temperature conditions. In the case of poultry enterprises supplying large volumes of products to urban consumers, waste recycling becomes a huge problem, and, in this case, pyrolysis and hydrothermal treatment become the main ways of solving serious environmental issues. Modern mobile pyrolysis equipment and flash pyrolysis technologies can contribute to a sharp reduction in capital costs for the organization and application of such technological solutions in practice. Alkaline treatment is presented as the simplest and most effective way to process small enough amounts of AW to obtain products ready for use. It may be interesting for enterprises producing such AW to diversify their processing methods, that is, to organize the parallel use of different methods of HF in order to obtain products with different practical purposes.

Most importantly, it is possible to manage the current situation by improving the quality of the HSs obtained, varying the processing methods of AW, using additives, and obtaining various biomimetic products with the best potential for practical application. A good scientific understanding of the current situation is very useful in this case.

## 6. Conclusions

The artificial HF of AW provides a solution to several tasks at once, reducing the volume of accumulated biowastes with N-containing compounds and yielding HS biomimetics capable of being used to overcome various environmental problems in the same ways (bioremediation, decontamination, fertilization, etc.) as natural HSs from peat and coal are used. Waste treatment methods are available for selection due to the use of different methods capable of converting the bioorganic compounds present in AW to various products with HSs of certain characteristics. The additives loaded in the reaction media with the treated AW can stimulate the formation of HSs with desirable chemical properties and increase the yields of the used processes. The characteristics of artificially obtained HSs from AW guarantee their use for agriculture or nature-like technologies, and in some cases, it can be even more attractive compared to natural HSs. The complete list of possible applications of artificial HSs has not yet been investigated, and it has notable potential for the further development of science in this area. However, the decision making on the choice of method obviously depends on the volume of accumulated wastes, the need for their rapid processing, and the availability of technical means that comply with the requirements for the technological conditions of HF (temperatures, pressure, etc.). In addition, the production of HSs as biomimetics is a result of the implementation of certain processes, and it is determined by the need for these HSs as well as availability of the processing sources themselves. For example, the use of HSs from thermo- or pyrolytic coal is available in any country engaged in agriculture, making such a process economically independent, whereas the use of similar HSs extracted from natural coal requires coal’s extraction, transportation, or purchase from the countries producing it. Thus, the information collected and summarized in this review clearly represents a pool of possible solutions for sources and methods of obtaining artificial HSs, the need for which increases annually. According to projections [132], the world market of HSs will increase by 11.8% and reach USD 1.1 billion by 2028.

## Figures and Tables

**Figure 1 biomimetics-08-00613-f001:**
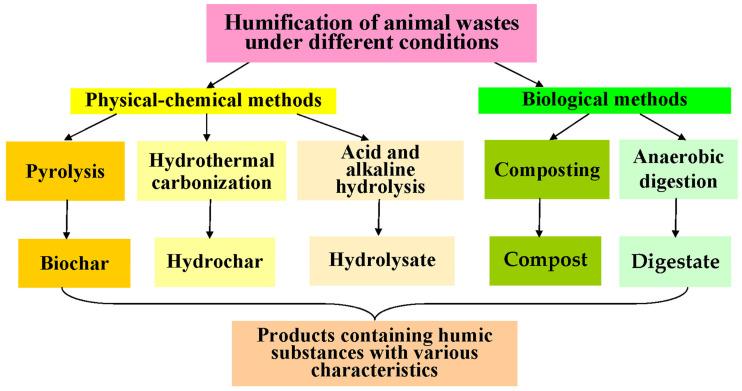
The scheme used in this review for a comparative analysis of the main HF processes of AW under different conditions and the HS resources obtained in them.

**Figure 2 biomimetics-08-00613-f002:**
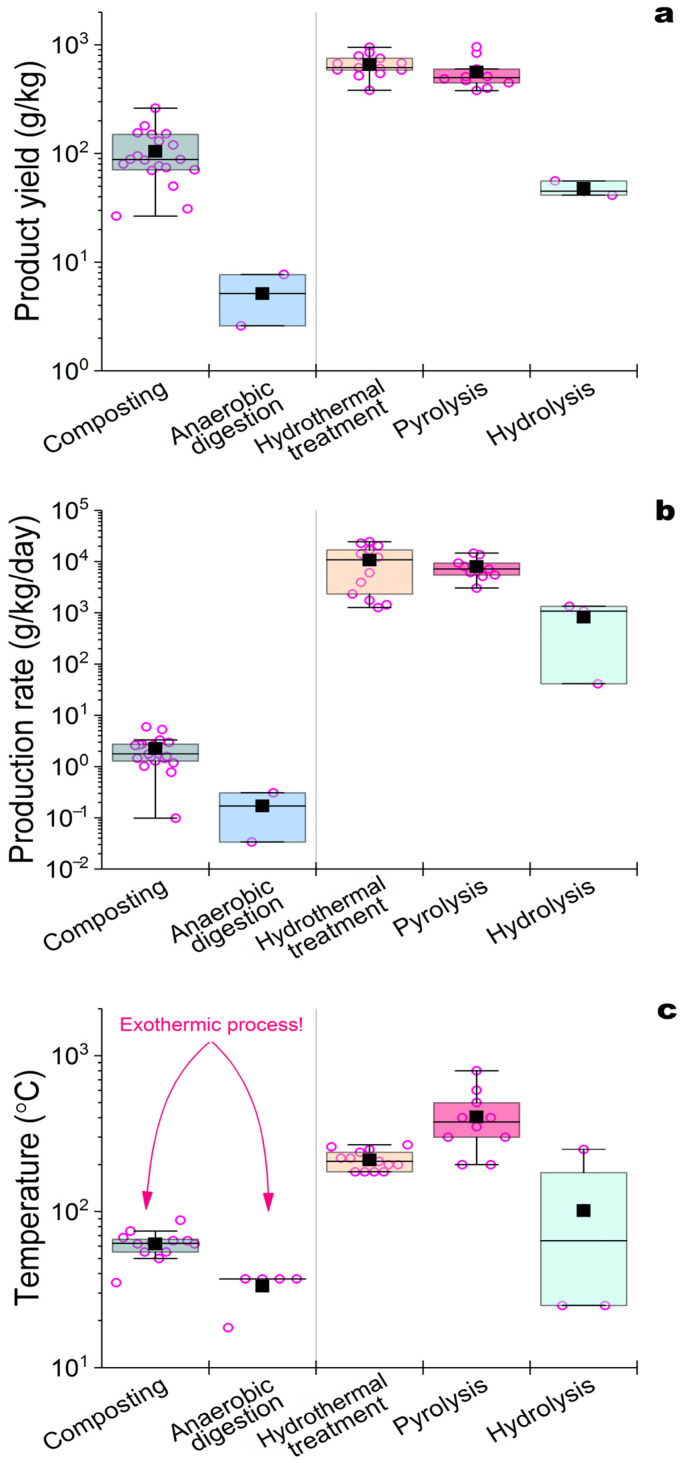
Box-plot visualization of HS yields (**a**), HS production rates (**b**), and temperatures (**c**) in various processes of HF AW presented in Table 1. Designations: ◯—individual data point; ∎—mean value; the box confines the interquartile range (25–75%) and is sectioned by segment at median value; the whiskers denote the value of 1.5 interquartile range. The rectangles of different colors correspond to certain HF processes indicated on the abscissa axis.

**Figure 3 biomimetics-08-00613-f003:**
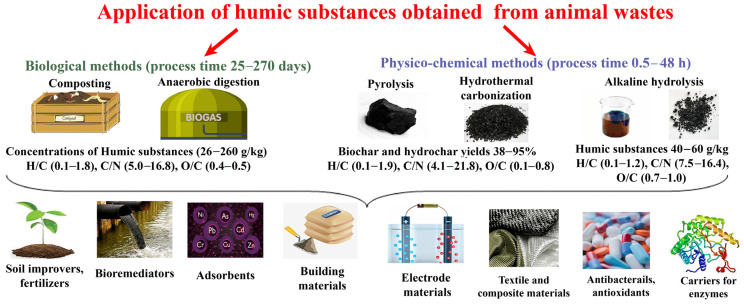
Properties of biomimetic HSs obtained via various methods and their practical potential.

**Table 1 biomimetics-08-00613-t001:** Annual amounts of animal waste produced in various countries.

Country/Reference	Animal Wastes [References]	AP *
USA	Dairy manure [11]	24,000
China	Livestock manure [12]	3800
	Chicken manure [13]	155.0
Brazil	Cattle manure [14]	1900
EU	Farm manure [15]	1200
France	Farm manure [16]	214.3
Germany	Farm manure [16]	175.7
The United Kingdom	Farm manure [16]	112.0
Spain	Farm manure [16]	108.3
Bangladesh	Cow manure [17]	102.6
Poland	Farm manure [16]	91.3
Italy	Farm manure [16]	89.4
India	Poultry manure [18]	38.0
Malaysia	Chicken manure [19]	23.1
Serbia	Farm manure [16]	18.6
Greece	Farm manure [16]	16.9
Belgorod Region, Russia	Total manure [20]	14.2
Turkey	Chicken manure [21]	11.0
Canary Islands	Livestock manure [22]	0.5
Malta	Farm manure [16]	0.3
South Africa	Cattle manure [23]	0.1

* AP—Annual production, million tons/year.

**Table 2 biomimetics-08-00613-t002:** Processing of AW with the production of HSs *.

Substrate (References)	Conditions/Additives	Products
**Composting**		
Dairy manure [28]	Thermal pretreatment (90 °C, 4 h), 60 days	Compost with 75.0–77.0 g of HS/kg
Cow dung and corn straw (ratio of 1:2) [29]	Addition of 2.5–5% (d.w.) FeSO_4_, 50 days	Compost with 109.8–129.9 g of HS/kg
Maize straw and chicken manure (ratio of 6:1) [30]	Addition of benzoic acid (5% d.w.) and soybean residue after oil extraction (15% d.w.), 62 days	Compost with 150.0 g of HS/kg
Dairy manure and sugarcane leaves and (ratio of 4:1) [31]	Two-step inoculation (0 and 9 days) by *Bacillus licheniformis*, *Aspergillus nidulans* and *A. oryzae* cells (ratio of 1:1:1 *w/w*/*w*)—2% d.w., 45 days	Compost with 70.0 g of HS/kg
Fresh dairy manure and sawdust (ratio of 3.5:1) [32]	Treatment with 0.2 M of H_2_O_2_ (0.5 L) and CuCl_2_ (0.5 g/kg of compost), 46 days	Compost with 151.9 g of HS/kg
Pig manure and sawdust (ratio of 2:1) [33]	Addition of Black Tourmaline—10% d.w., 42 days	Compost with 50.2 g HA/kg and 24.0 g FA/kg
Dairy manure and bagasse pith (ratio of 3:1) [34]	Addition of H_2_O_2_ (2.14 mmol/kg) and ascorbic acid (3.57 mmol/kg of the d.w.), 34 days	Compost with 180.0 g of HS/kg
Chicken manure and rice husk (ratio of 6.7:1) [35]	Hyper thermophilic pretreatment (≥80 °C) for 1–9 days and total process for 44 days	Compost with 65% HS of TS (according to calculations ~260 g of HS/kg)
Pig manure and rice straw (C/N = 25) [36]	Hyper thermophilic pretreatment (90 °C, 4 h), 60 days	Compost with 87.8 g of HS/kg
Chicken manure and corn straw (C/N = 20) [37]	Addition of malonic acid (0.5%), MnO_2_ (0.5% d.w.), or their combination, 60 days	Compost with 75.0–87.0 g of HS/kg
Chicken manure, sawdust and urea (C/N = 30) [38]	Addition of 0.1% adenosine triphosphate or 0.5% malonic acid (d.w.), 49 days	Compost with 40.0–50.0 g of HS/kg
Digestates and chicken manure [39]	Without additives, 60 days	Compost with 90.0–95.0 g of HS/kg
Swine manure and corn stalk (ratio of 6:1) [40]	Addition of 1.0% (*v*/*w*) *Acinetobacter pittii*, *Bacillus subtilis, B. altitudinis* (ratio of 1:2:1 *v*/*v*), 32 days	Compost with 88.1 g of HS/kg
Cattle manure (6.7–30% dry basis), rice straw (21.7–31.7%), biogas residue (30–70%), food waste (8.3%) [12]	Without additives, 30 days	Compost with 75.0–88.5 g of HS/kg
Dairy manure and bagasse [41]	Addition of 10% Red mud (d.w.), pH 8.7, 45 days	Compost with 115.0–120.0 g of HS/kg
Cow manure and sugar cane straw (ratio of 5:1) [42]	Addition of 5% biochar from wood obtained via high-temperature gasification (400–550 °C), 40 days	Compost with 29.0–31.0 g of HS/kg
Chicken manure and rice hulls (C/N = 25) [43]	Addition of lignite (15% *w*/*w*), 55 days	Compost with 80.2 g of HS/kg
Chicken manure and rice straw (C/N =25–30) [44]	Addition of 7.5% montmorillonite (*w*/*w*) and pretreatment at 550 °C, 60 days	Compost with 67.0–71 g of HS/kg
Chicken manure and spent mushroom substrate (ratio of 1:1.2) [45]	Addition of Garden waste (15% fresh weight), 60 days	Compost with 145.0–155.0 g of HS/kg
Horse manure (C/N = 33) [46]	Vermicomposting (10 g of earthworms (*Eisenia andrei*)/kg), 35 °C, 6–9 months	Compost with 26.0–26.6 g of HA/kg
**Anaerobic digestion**		
Chicken manure [47]	37 °C, 10.0% of TS, and 7.9% VS, 40 days	Digestate—relative content of HLC (34%) and FLC (6%). HS yield was not controlled
Chicken manure [48]	37 °C, 10.0% TS, and 7.9% VS, 25 days	Digestate—7.7 g HA/L
Turkey manure [49]	37 °C, 51.2% (*w*/*w* wet basis) TS, and 71.5% (*w*/*w* dry basis) VS, OLR—0.5–2.5 kg VS/m^3^ per day, 77 days	Content HS in liquid fraction of the effluent and entire effluent (with digestate)—2.36 (2.32 HA, 0.04 FA) and 2.6 (2.04 HA, 0.60 FA) g/L
Sheep bedding and cattle manure [50]	18 ± 4 °C; sheep-bedding-to-cattle-manure ratios of 0:100, 25:75, 50:50, 75:25, and 100:0; final content of TS –5%; 5 months	Digestate with HA/FA—1.3–3.0. HS yield was not controlled
Pig manure [51]	Hydrothermal pretreatment (70–170 °C, 0.5 h), 37 °C, 30 days	Digestate with HLC and FLC in amounts of 58.0–65.9 and 35.5–42.0%, respectively. HS yield was not controlled.
**Hydrothermal carbonization**		
Dried swine manure [52]	180 °C, 1 MPa, 15wt.% CaO, 10 h	HCmy—75.2%
Dried poultry litter [53]	180 °C, 1 MPa, 1 h	HCmy—60.4%
Dried poultry litter [54]	250 °C, 4–5 MPa, H_2_SO_4_ (pH 2.0), 2 h	HCmy—38.1%
Dry swine and chicken manure [55]	240 °C, 3–4MPa, 10 h	HCmy—54.6%
Dried swine manure with cellulose [56]	210 °C, 2 MPa, 5 h	HCmy—52.0%
Dried swine manure with sawdust [57]	220 °C, 2–3 MPa, 10 h	HCmy—61.8%
Dried pig manure [58]	180 °C, 1 MPa, 1–1.5 g KOH per 100 g manure, 1 h	HCmy—79.0%
Dried swine manure [59]	200 °C, 2 MPa, 30 min	HCmy—58.7%
Chicken litter [60]	220 °C, 2–3 MPa, 20 min	HCmy—68.0%
Air-dried pig manure [61]	200 °C, 2 MPa, 2 h	HCmy—58.8%
Poultry and swine manure; dairy and beef cattle manure; broiler and layer chicken litter [62]	180 °C, 1 MPa, 1 h	HCmy—67.3%
Mixture of chicken manure with sawdust [63]	260 °C, 40 min	Biochar yield—95.1%
Dewatered poultry sludge [64]	268 °C, 47 min	Biochar yield—85.0%
**Pyrolysis**		
Dried pig manure [58]	200 °C, 1 h	Biochar yield—40.0%
Poultry litter [65]	Wet torrefaction pretreatment (300 °C), 600 or 800 °C, supercritical CO_2,_ 1.5–2 h	Biochar yield—51.2%
Chicken litter [60]	400 °C, 20 min	Biochar yield—38.0%
Pre-dried broiler manure [66]	350 °C, 1 h	Biochar yield—47.0%
Dried poultry litter [67]	500 °C, Mixed with H_3_PO_4_ and MgO (biomass:H_3_PO_4_ ratio = 1:0.5 (*w*/*w*), molar P:Mg ratio—1:1), 2 h	Biochar yield—60.0%
Air-dried pig manure [61]	300 °C, 1 h	Biochar yield—84.0%
Dried digested cattle manure [68]	600 °C, 30 min	Biochar yield—44.8%
Poultry and swine manure, dairy and beef cattle manure, broiler and layer chicken litter [62]	400 °C, 1 h	Biochar yield—51.0%
Dried goat manure [69]	300 °C, 30 min	Biochar yield—48.6%
Air-dried poultry manure [70]	200 °C, 4 h	Biochar yield—95.8%
**Hydrolysis**		
Air-dried sheep or cow manures [71]	Acid hydrolysis (0.1–1.0 N HCl or H_2_SO_4_) at 105 °C and extraction (1N KOH), 1 h	HS yield—45 g/kg (sheep waste) and 56 g/kg (cow waste)
Air-dried poultry manure [72]	25 °C, 0.1 N NaOH, 24 h	HA—28.1 g/kg. FA—13.3 g/kg
Farmyard manure [73]	25 °C, 0.1 M NaOH, 450 rpm, 48 h	HA yield—10%
Fresh chicken manure [74]	Subcritical water extraction (230–250 °C, 6 MPa)	Liquid phase with 31.0 g of HA/kg and 20 g of FA/kg

* VS—volatile solids, TS—total solids, HCmy—hydrochar maximal yield, OLR—organic loading rate, HLC—humic-like compounds, and FLC—fulvic-like compounds.

**Table 3 biomimetics-08-00613-t003:** Combination of AFPs with different antifungal agents.

Sample of Waste [References]	Chemical Elements (%)					Ratios		
	C	H	N	O	S	H/C	C/N	O/C
**Composting**								
Chicken manure and rice husk [35]	41.10	3.40 ^a^	5.70 ^a^	n/d	n/d	0.08	7.14	n/d
Horse manure [46]	38.42 ^a^	42.36 ^a^	2.29 ^a^	17.19 ^a^	n/d	1.10 ^b^	16.76 ^b^	0.47 ^b^
**Anaerobic digestion**								
Chicken manure [48]	n/d	n/d	n/d	n/d	n/d	1.78 ^b^	5.01 ^b^	n/d
**Hydrothermal carbonization**								
Dried swine manure [52]	35.09	4.64	1.97	26.65	n/d	0.13	17.8	0.76
Dried poultry liter [53]	33.61	3.91	1.95	20.84	0.31	0.12	17.2	0.62
Dried poultry liter [54]	56.40	4.99	5.13	7.78	1.22	0.09	11.0	0.14
Dried swine manure [55]	40.61	4.15	2.11	11.72	0.18	0.10	19.2	0.29
Dried poultry manure [55]	28.44	2.84	2.05	5.65	0.25	0.10	13.9	0.20
Dried swine manure [56]	40.42	3.71	1.94	18.10	0.18	1.10 ^b^	20.8	0.33 ^b^
Dried swine manure with sawdust [52]	40.85	6.30	3.73	31.30	0.42	1.55 ^b^	11.0	0.57 ^b^
Poultry liter [60]	37.5	n/d	8.01	n/d	n/d	n/d	4.7	n/d
Dewatered poultry sludge [64]	53.43	8.17	3.67	11.24	0.52	1.86 ^b^	14.6	0.16 ^b^
Air-dried pig manure [61]	33.77	4.22	2.49	14.96	0.55	1.50 ^b^	13.6	0.33 ^b^
Swine manure Zhou [62]	35.96	4.36	2.02	22.30	0.54	0.12	17.8	0.62
Daily cattle manure [62]	43.63	5.27	2.17	26.92	0.58	0.12	20.1	0.62
Beef cattle manure [62]	38.79	4.33	1.78	24.10	0.48	0.11	21.8	0.62
Broiler liter [62]	38.19	4.53	3.45	23.57	0.65	0.12	11.1	0.62
Layer chicken liter [62]	39.58	5.02	2.09	22.04	0.62	0.13	18.9	0.56
**Pyrolysis**								
Dried poultry manure [67]	43.30	2.15	n/d	n/d	n/d	0.05 ^b^	n/d	n/d
Air-dried pig manure [61]	29.04	1.41	1.36	0.29	4.82	0.58 ^b^	21.6	0.12 ^b^
Air-dried poultry manure [61]	39.70	5.62	3.53	42.3	n/d	0.14	11.2	1.07
Dried digested dairy cattle manure [68]	39.60	0.85	1.84	n/d	0.94	0.02	21.5	n/d
Swine manure [62]	34.89	1.96	2.44	10.16	0.51	0.06	14.3	0.29
Daily cattle manure [62]	42.27	2.41	2.46	11.39	0.57	0.06	17.2	0.27
Beef cattle manure [62]	40.55	2.09	2.04	11.39	0.45	0.05	19.9	0.28
Broiler liter [62]	37.59	2.18	4.57	6.13	1.06	0.06	8.23	0.16
Layer chicken liter [62]	35.39	1.98	2.52	7.76	0.88	0.06	14.0	0.22
**Alkaline hydrolysis**								
Air-dried poultry manure [72]	45.06	4.08	6.01	44.85	n/d	0.09 ^b^	7.49	0.99 ^b^
Farmyard manure [73]	53.10	5.45	3.24	37.63	0.58	0.10 ^b^	16.39 ^b^	0.71 ^b^
**Natural HS extracted from different environmental sources (for comparison)**								
HA from peat [93]	40.1	4.2	2.5	n/d	2.2	0.10	16.0	n/d
HA from peat [94]	52.25	4.51	2.59	n/d	0.77	1.03 ^b^	20.2	0.57 ^b^
HA from peat [94]	56.34	5.71	2.34	n/d	0.88	1.20 ^b^	24.1	0.45 ^b^
HA from raw lignite coal [95]	72.20	4.44	1.97	18.07	3.31	0.06 ^b^	36.6	0.25 ^b^
HA from native bituminous coal [95]	56.20	10.99	3.07	18.59	11.15	0.19 ^b^	18.3	0.18 ^b^

^a^ Concentration of HA; ^b^ data from publications. Ratios without this index were calculated for this review based on the data presented in the cited references; n/d—no data.

**Table 4 biomimetics-08-00613-t004:** Application of artificial and natural HS resources.

Product with HSs	Application	Characteristics
**HSs from animal wastes**		
Chicken manure biochar [99]	Adsorbent for the removal of phenol and 2,4-dinitrophenol from wastewater	Maximum adsorption capacity: 106.2 mg/g phenol, 148.6 mg/g 2,4-dinitrophenol
Chicken manure biochar [100]	Remediation of metals from water and soil	Removal efficiency: 98% of Pb^2+^; 42% of Zn^2+^
Swine manure biochar [101]	Adsorption of U(VI)	Maximum adsorption capacity: 221.4 mg/g
Swine manure hydrochar modified with manganese ferrite (MnFe_2_O_4_) nanoparticles [102]	Removal of chlortetracycline and Cd (II) from water	Maximum adsorption capacity: 753.0 mg/g (chlortetracycline), 62.2 mg/g (Cd (II)),
Cattle manure hydrochar [103]	Soil conditioner	Hydrochar improves total soil phosphorus (by 6.8–18.9%), soil organic carbon (by 8.2%), dissolved organic carbon (by 18.7%), rice yield (by 36.9%)
Swine manure hydrochar [104]	Removal of metal from aqueous solutions	Maximum adsorption capacity (mg/g): 81.1 (Cd II), 13.1 (Sb III)
Hydrochar made from digestate of manure [105]	Soil amendment	Increase in soil pH (from 7.0 to 7.4), cation exchange capacity (from 11.5 to 12.6 meq/100 g soil), soil organic matter (from 2.4 to 2.8%), and P, K, Ca, and Mg content. Twofold increase in dry weights of roots, leaves, and plant *Lactuca sativa*
Compost made from farmyard manure with addition of biochar [106]	Soil amendment	Increase in growth, yield, and chlorophyll content and decrease in Cd content in wheat tissues
HSs extracted from compost containing cow manure [107]	Biosurfactant	Percent of metal removal: Cu—17%, Pb—35%, Zn 8%, Cd—38% and Cr—0.6%
Hydrochar from digestate of cow manure and corn straw [108]	Additive to AD	Enhancement of CH_4_ yield—34%
Chicken manure biochar [109]	Additive to composting mass	Decrease in emissions of N_2_O, CH_4_, and NH_3_ by 27.4%, 55.9%, and 56.9%, respectively
**HSs from other types of wastes**		
HA from compost [110]	Treatment of diesel-contaminated soil	Diesel removal—89.4%
HA from vermicompost [111]	Adsorption of aflatoxin B_1_ (AFB_1_) from maize-soybean meal for broiler chickens (100 µg AFB_1_/kg)	Improved adsorption—99.7%
Mixture of bamboo biochar and pig manure [112]	Soil remediation	Soil treatment with biochar–pig manure increased concentration of arbuscular mycorrhizal fungi
Biochar obtained through the pyrolysis of bamboo and rice husk [113]	Additive to composting mass	Improvement of organic matter decomposition, enhanced HA concentration (>80 g/kg), reduced volatilization of NH_3_ and N_2_O (40%)
Wheat straw biochar [114]	Additive to composting mass	Notable prolongation of thermophilic period of pig manure composting with stabilization of bacteria richness.
Hydrochar from olive mill waste and cellulose [115]	Enzyme immobilization	Absorption immobilization of enzyme—20–30%
Corn straw hydrochar [116]	Electrode material	Mass-specific capacitance—98 F/g. Power density—9500 W/kg. Energy density—77 W h/kg at 20 A/g
HS extracted from composted artichoke residues [117]	Antibacterial agent and antioxidant	Minimal inhibitory concentrations (mg/L) against bacterial cells in concentration of 5 × 10^5^ CFU/mL: against *Staphylococcus aureus*—1.2, *Pseudomonas aeruginosa*—1.8, *Enterococcus faecalis*—2.0, *Escherichia coli*—1.7, *Klebsiella pneumoniae*—2.3. Antioxidant activity (expressed as gallic acid equivalents)—150 mol/g.

## Data Availability

Not applicable.

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
