# Peer review of "Artificial Humic Substances as Biomimetics of Natural Analogues: Production, Characteristics and Preferences Regarding Their Use"

_biomimetics, 2023, doi:10.3390/biomimetics8080613_

Round 1

Reviewer 1 Report

Comments and Suggestions for Authors

Humidification is one of the most promising ways to treat animal waste while producing humic substances for multiple uses. Since not many review papers were published on this topic, the manuscript shows valuable information such as different types of humidification methods, the related humic substances with different characteristics, and the additives used to increase the effectiveness of humidification. However, minor revisions were suggested to improve the quality of the manuscript before it could be accepted to be published in Biomimetics. Please find below my detailed comments:

  1. In the opinion of the authors, which humidification method should be prioritized in consideration of production time, funding, humidification rate, and the quality of humic substances?
  2. In the opinion of the authors, which type of manure is convenient for humidification and gives products (compost, digestate, biochar, and humic substances) the highest quality?
  3. Which preferred additives should be used for each humidification method?
  4. The abstract should be written more concisely, while the conclusions should be extended with more findings and opinions.
  5. Good publications on catalysts for enhanced humidification in composting (Journal of Cleaner Production (2023), 402, 136832 [DOI: 10.1016/j.jclepro.2023.136832]) and advances in the characterization of humic substances (Membrane Water Treatment (2020), 11, 4, 257-274 [DOI: 10.12989/mwt.2020.11.4.257]) should be referred to to increase the quality of the manuscript.
Comments on the Quality of English Language

Some typos were found throughout the text.

Author Response

Dear Reviewer,

The authors are grateful to you for the suggestions allowing us the improving of our manuscript.

Please, see our comments to your remarks and the revised text of the paper:

Comments and Suggestions for Authors

  1. In the opinion of the authors, which humidification method should be prioritized in consideration of production time, funding, humidification rate, and the quality of humic substances?

Response: The authors are grateful for the question because it demonstrates the reality of practical applications of the Review’s information. By treating the information coming from various literature sources we tried to estimate the obtaining of humic substances from animal wastes by using different methods. As you know, we estimated velocity of the processes, product yields, temperature of the treatments, and characteristic of obtained humic substances. The analysis of these data was initially performed after the Figure 2 and before Figure 3 in the text of Review (please, see the part 2.2 and part 4 respectively). Commonly, the composting is the most popular method of organic matter obtaining (humification), but it is not the best method among others (it is very long and nonproductive). It can be used for the treatment of relatively low amounts of animal wastes with good outside temperature conditions. In the case of poultry farms supplying large volumes of products to urban consumers, waste recycling becomes a huge problem, and in this case pyrolysis and hydrothermal treatment become the main ways to solve serious environmental problems. Modern mobile pyrolysis plants and flash pyrolysis technologies can contribute to a sharp reduction in capital costs for the organization and application of such technological solutions in practice. Alkaline treatment is presented as the simplest and most effective way to process, again, small amounts of waste to produce products ready for use. It may be interesting for enterprises with such waste to diversify their processing methods, that is, to organize the parallel use of different methods of humification in order to obtain products with different practical purposes.

This response was added to the text of the Review before the Conclusion.

  1. In the opinion of the authors, which type of manure is convenient for humidification and gives products (compost, digestate, biochar, and humic substances) the highest quality?

We tried to estimate the situation with types of animal wastes in the Review. In was very interesting for ourselves, because we did not find similar reviews in literature. There are a lot of papers about the treatments of lignocellulose wastes but not a lot of animal wastes coming from poultry industry. The main response concerning the authors’ opinion is following: there are no good or worst animal wastes, there are problems with them and we have options for the problem solutions. We can manage the situation, improving the quality of the obtained HS, varying processes for AW treatment, using additives, obtaining different products with their best potential for practical application. Good scientific understanding of the current situation is very helpful.

This response was partially added to the text of the Review before the Conclusion.

  1. Which preferred additives should be used for each humidification method?

It is impossible to unambiguously determine today which additives are the best for a particular method of waste humification, and there are no universal solutions to the problem of improving the quality of the HS obtained and the efficiency of the HF processes themselves. That is why such studies remain relevant and continue to be conducted. Perhaps combinations of already known additives can make additional changes to the discussed characteristics of HS and processes.

This response was partially added to the text of the Review before the Figure 3.

  1. The abstract should be written more concisely, while the conclusions should be extended with more findings and opinions.

The abstract was modified and the text was added to the Conclusion in accordance with the recommendations of the Reviewer.

  1. Good publications on catalysts for enhanced humidification in composting (Journal of Cleaner Production (2023), 402, 136832 [DOI: 10.1016/j.jclepro.2023.136832]) and advances in the characterization of humic substances (Membrane Water Treatment (2020), 11, 4, 257-274 [DOI: 10.12989/mwt.2020.11.4.257]) should be referred to to increase the quality of the manuscript.

The first one good publication was added to the text of Review accordingly to the recommendation of the Reviewer. The second publication was not introduced to the text of the Review because it deals with methods of estimations of humic substances in the flow within the process of water treatments. This question is not discussed in the text, but the authors are grateful to the Reviewer for this reference. It can be used by the authors in their other publications. Since the insertions was made, the numeration of references was changed in the whole text.

Comments on the Quality of English Language: Some typos were found throughout the text.

  The authors are grateful to the Reviewer for this comment, and the text was additionally checked for typos.

With kind regards,

The authors of the manuscript

Reviewer 2 Report

Comments and Suggestions for Authors

This is a review which needs major revision.

First, the possible application of commercial humic substances should be compared to annual in situ formation of humic substances, worldwide. With this calculation it is possible to get informations about the quantitative relevance of commercail humic applications.

Second, humification or formation of humic substances should be defined. For this purpose the recent discussion of the relevance of humic substances should be briefly discussed (e.g. Lehmann and Kleber, 2015, Nature, 528, 61 ff.; Gerke, 2018, Agronomy, 8:76; Hayes and Swift, 2020, Adv. Agron., 163, 1ff.). Unless the authors prove their definition, pyrolysis, hydrthermal carbonization are no procedures to produce humic substances. With alkaline hydrolysis, do you mean alkaline extraction of humic substances??

Some minor comments:

L 83 HS precursors, include aromatic proecursors.

Table 1 Add unit.

The results of Y. Chen et al. (e.g. Inbar et al., 1990, Soil Sci. Soc. Am. 54, 1316 ff.) should be mentioned.

Comments on the Quality of English Language

see above

Author Response

Dear Reviewer,

The authors are grateful to you for the suggestions allowing us the improving of our manuscript.

Please, see our comments to your remarks and the revised text of the paper:

Comments and Suggestions for Authors

First, the possible application of commercial humic substances should be compared to annual in situ formation of humic substances, worldwide. With this calculation it is possible to get informations about the quantitative relevance of commercail humic applications.

The sources of humic substances worldwide are annually all bioorganic sources, which are grown plants, food and paper waste from municipal farms, waste from agriculture and the processing and food industry. Theoretically, these wastes can be converted into final substances containing different concentrations of humic substances with different chemical characteristics. We cannot estimate these volumes on a worldwide scale. We tried to do this using the example of bird droppings by collecting and presenting such data in Table 1. We believe that in this way we have confirmed the relevance of the problem discussed in the Review.

As for the commercial sale of humic substances and the need for them, we have inserted a link to the website in the text (reference #129), where it is indicated that such commercial interest in humic substances exists and the need for them is expected to increase by 11.8% worldwide by 2028 and reach $1.1 billion [https://www.gminsights.com/industry-analysis/humic-acid-market]. The commercial significance of the use of humic substances is determined by the positive contribution that can be obtained from their use in agriculture, remediation processes etc.

At the same time, it should be noted that not all biological resources and the humic acids contained in them can be commercially used. The reason is that they will have constantly varying characteristics due to the variability in the composition of the initial components to be converted to humic acids. Such variants of humic acids are not suitable for scalable commercialization. It is animal husbandry waste, where the composition of waste is relatively constant due to the constancy of the composition of the feed used, which are additionally checked for the presence of heavy metals and a number of toxins, can serve as a good raw material for its processing into humic substances for commercial purposes.

Second, humification or formation of humic substances should be defined. For this purpose, the recent discussion of the relevance of humic substances should be briefly discussed (e.g. Lehmann and Kleber, 2015, Nature, 528, 61 ff.; Gerke, 2018, Agronomy, 8:76; Hayes and Swift, 2020, Adv. Agron., 163, 1ff.).

The authors are grateful to the Reviewer for this comment. The definition of humification was added to the text at the beginning of the article. Several references including those recommended by the Reviewer, were additionally introduced to the text. So, the general numeration of references. The discussion of chemical content of various HS and the content of chemical elements typical for them is performed in the part 3 and 4 of the text.

  1. Lehmann, J.; Kleber, M. The contentious nature of soil organic matter. Nature. 2015, 528 (7580), 60-68. doi:10.1038/nature16069
  2. Gerke, J. Concepts and misconceptions of humic substances as the stable part of soil organic matter: A review. Agronomy 2018, 8, 76. doi:10.3390/agronomy8050076
  3. Wang, M.; Li, Y.; Peng, H.; Wang, J.; Li, Q.; Li, P.; Fan, J.; Liu, S.; Zheng, G. Review: Biotic and abiotic approaches to artificial humic acids production. Sust. Energ. Rev. 2023, 187, 113771. doi:10.1016/j.rser.2023.113771.

Unless the authors prove their definition, pyrolysis, hydrthermal carbonization are no procedures to produce humic substances.

The appearance of humic substances formed during the processes discussed in the Review, as well as their chemical composition, are confirmed by the authors of numerous articles themselves, which were reviewed and presented in the form of an overview. Evidence that humic substances are formed during various processes has been obtained by the authors of numerous studies themselves. Their results have been published, including in referred well-known journals, and we have no doubts, so we used them as sources of information and references. The very relevance of the artificial production of humic substances has long been beyond doubt, therefore, researchers in different countries of the world are turning to ways to solve the problem of bioorganic waste disposal, looking for the best solutions, understanding exactly how to influence processes, manage them and use the results obtained in the best way.

With alkaline hydrolysis, do you mean alkaline extraction of humic substances??

No, this is not an extraction at all. There are practically no (!) humic substances in the initial manure, there is nothing to extract, unlike coal and peat, where these substances are already present in the form formed over many years, and they can be extracted. In the case of manure, the following processes occur under the action of alkaline agents: hydrolysis of residual lipids, hydrolysis of proteins to form peptides and amino acids, hydrolysis of residual carbohydrates to oligo- and monosaccharides (since grain is initially present in poultry feed, which means mainly starch). All these processes follow the usual nucleophilic mechanism. In the presence of alkaline agents, as is well known, the formation of di-, tri- and polyglycols from glucose occurs, which is formed during the decomposition of residual amounts of starch (it is present in feed grains used for animals). In addition, alkaline hydrolysis of nucleotides (DNA and RNA of plant cells) occurs, the resulting products are also present in the reaction medium. Due to the presence of alkaline agents in the reaction medium and water in the manure, unlike coal and peat, which, as a rule, have low humidity, the hydration process of many organic compounds occurs. These processes occur with an increase in the temperature of the reaction medium, which accelerates the hydrolytic processes in question. In addition, the presence of oxygen in the medium (which is practically absent in coal and peat) in an alkaline medium, multiple oxidative processes occur, and an alkaline medium contributes to this, since many substances involved in these processes retain their solubility. It should be noted that it is the ability of alkaline agents to catalyze the hydrolysis of many organic substances that allows them to be used not for extraction, but for dissolution and hydrolysis, including serious organic pollutants. All products formed under alkaline conditions can further participate in the formation of supramolecular complexes due to multiple weak chemical interactions, forming just humic substances.

Some minor comments:

L 83 HS precursors, include aromatic proecursors. - It was corrected. Thank you.

Table 1 Add unit. - It was corrected. Thank you.

The results of Y. Chen et al. (e.g. Inbar et al., 1990, Soil Sci. Soc. Am. 54, 1316 ff.) should be mentioned.

Sorry, we did not find the same article of the authors recommended by the Reviewer, but we added another their publication as reference #4: Chen, Y., Inbar, Y., Chefetz, B., Hadar, Y. (1997). Composting and recycling of organic wastes. In: Rosen, D., Tel-Or, E., Hadar, Y., Chen, Y. (eds) Modern Agriculture and the Environment. Developments in Plant and Soil Sciences, vol 71. Springer, Dordrecht. https://doi.org/10.1007/978-94-011-5418-5_28

With kind regards,

The authors of the manuscript

Round 2

Reviewer 2 Report

Comments and Suggestions for Authors

The authors have improved their paper. I recommend further clarification on the definition of humic substances produced by pyrolysis. Humic substances in soils consist of aliphatic and aromatic compounds, the latter being condensed and non-condensed aromatics. However pyrolysis of organic matter is expected to mainly yield condensed compounds. This should be mentioned in the paper. Condensed aromatics in humic substances have been falsely attributed to pyrogeneous carbon as it is now evident by the work of the group of Pat Hatcher.

Author Response

Dear Reviewer,

The authors are grateful to you for the suggestions allowing us the improving of our manuscript.

Please, see our comments to your remarks and the revised text of our review.

Comments and Suggestions for Authors

The authors have improved their paper. I recommend further clarification on the definition of humic substances produced by pyrolysis. Humic substances in soils consist of aliphatic and aromatic compounds, the latter being condensed and non-condensed aromatics. However pyrolysis of organic matter is expected to mainly yield condensed compounds. This should be mentioned in the paper. Condensed aromatics in humic substances have been falsely attributed to pyrogeneous carbon as it is now evident by the work of the group of Pat Hatcher.

Response

While preparing a response to the recommendations of the Reviewer, we accepted all our corrections made in the text at the previous stage of work on the article, so that it could be seen what additions were made to the article this time.

We have great respect for the professionalism of the Reviewer, who helps us improve our review, and we are very grateful to the Expert for his attentive attitude to the discussion of the composition of humic substances (HS), which are formed as a result of the application of various processes used for the treatment of organic wastes. Our initial desire to collect information on the treatment of animal wastes, which is a huge and extremely serious environmental problem, unlike plant waste, in products with HS stimulated us to prepare this review. We summarized and analyzed how HS differ in their compositions, which can be obtained by various methods of processing animal excrements, and how close or far they are from natural variants, as well as among themselves. In this regard, we would like to draw the attention of the Reviewer to the fact that in those raw materials discussed in the work of Patrick Hatcher, recommended by the Reviewer in connection with their pyrolysis and the production of HS with condensed aromatic compounds, plant wastes are present, and the resulting pyrolysis products are obviously different from those that we mainly discuss in our review.

Since we could not completely avoid discussing plant biomass in the review when obtaining HS from animal wastes, since animal excrements often contain residues or impurities of lignocellulose raw materials, we initially included these references in the Tables in the review, and now we have added a reference to the work of reputable Patrick Hatcher in our review [DiDonato, N.; Chen, N.H.; Waggoner, D.; Hatcher, P.G. Potential origin and formation for molecular components of humic acids in soils, Geochimica et Cosmochimica Acta, 2106, 178, 210-222. https://doi.org/10.1016/j.gca.2016.01.013.]. At the same time, in an article by Patrick Hatcher et al. lignin is clearly indicated as the lignin as the primary source of soil organic matter and the main reason for the appearance of condensed aromatic compounds in pyrolysis of plant wastes. In case of manure there are no so much amounts of lignin, and moreover, in case of bird manure it is absent at all because the starch-containing feeding (grains) is used for these animals. However, the residues of plant litter (bedding) for birds can contain the lignin in some concentrations, and the aromatic compounds appeared in the content of HS obtained by pyrolysis of AW. This information is confirmed by another interesting article [Jin, J.; Sun, K.; Yang, Y.; Wang, Z.; Han, L.; Wang, X.; Wu, F.; Xing, B. Comparison between soil- and biochar-derived humic acids: composition, conformation, and phenanthrene sorption. Environ. Sci. Technol. 2018 52 (4), 1880-1888. https://doi.org/10.1021/acs.est.7b04999.], which was also added to the review, when we discussed the compositions of HS formed during pyrolysis of animal waste and their composting, since this determines the possibility and effectiveness of their practical application, for example, as sorbents.

At the same time the amounts of aromatic compounds in the composition of HS from AW depends on the temperature and duration of pyrolysis [Tomczyk, A.; SokoÅ‚owska, Z.; Boguta, P. Biochar physicochemical properties: pyrolysis temperature and feedstock kind effects. Rev. Environ. Sci. Biotechnol. 2020, 19, 191–215. doi:10.1007/s11157-020-09523-3]. The conditions can be varied, and the amounts of aromatic compounds can be increased targetly, for example, for the further application of the HS as sorbent of hydrophobic compounds like phenanthrene in soil. These additions were made to Part 4 of the review.

The numeration of the references was changed in the text after the additions.

With kind regards,

The authors of the manuscript
